# The Winter Habitat Selection of Red Deer (*Cervus elaphus*) Based on a Multi-Scale Model

**DOI:** 10.3390/ani10122454

**Published:** 2020-12-21

**Authors:** Yue Sun, Yanze Yu, Jinhao Guo, Minghai Zhang

**Affiliations:** 1College of Wildlife and Nature Reserve, Northeast Forestry University, Harbin 150040, China; sy1028sy@163.com (Y.S.); guojinhao19960206@126.com (J.G.); 2Heilongjiang Academy of Forestry, Harbin 150081, China; xiaoyu20090916@163.com

**Keywords:** red deer, habitat selection, multi-scale, ungulata

## Abstract

**Simple Summary:**

Most wildlife habitat studies are yet to adopt a multi-scale framework. Our study of the winter habitat selection of red deer (*Cervus elaphus*) is based on a multi-scale model and shows the importance of taking different scales into account when investigating habitat selection. Our approach captures a wide spectrum of ecological relationships of a population, results in effective conservation planning, and is readily applicable to other species of wildlife. Efficacy of future habitat selection studies will benefit by taking a multi-scale approach. In addition to potentially providing increased explanatory power and predictive capacity, multi-scale habitat models enhance our understanding of the scales at which species respond to their environment, which is critical knowledge required to implement effective conservation and management strategies.

**Abstract:**

Single-scale frameworks are often used to analyze the habitat selections of species. Research on habitat selection can be significantly improved using multi-scale models that enable greater in-depth analyses of the scale dependence between species and specific environmental factors. In this study, the winter habitat selection of red deer in the Gogostaihanwula Nature Reserve, Inner Mongolia, was studied using a multi-scale model. Each selected covariate was included in multi-scale models at their “characteristic scale”, and we used an all subsets approach and model selection framework to assess habitat selection. The results showed that: (1) Univariate logistic regression analysis showed that the response scale of red deer to environmental factors was different among different covariate. The optimal scale of the single covariate was 800–3200 m, slope (SLP), altitude (ELE), and ratio of deciduous broad-leaved forests were 800 m in large scale, except that the farmland ratio was 200 m in fine scale. The optimal scale of road density and grassland ratio is both 1600 m, and the optimal scale of net forest production capacity is 3200 m; (2) distance to forest edges, distance to cement roads, distance to villages, altitude, distance to all road, and slope of the region were the most important factors affecting winter habitat selection. The outcomes of this study indicate that future studies on the effectiveness of habitat selections will benefit from multi-scale models. In addition to increasing interpretive and predictive capabilities, multi-scale habitat selection models enhance our understanding of how species respond to their environments and contribute to the formulation of effective conservation and management strategies for ungulata.

## 1. Introduction

Examining habitat selection is one way to assess the importance of habitat to species conservation, but making such assessments is not always straightforward, even for well-studied species. We see two main obstructions to understanding habitat selection: first, the conceptual hurdles that obscure our knowledge of habitat selection and its underlying dynamics at multiple scales; and second, the practical limitations on sampling that stem, in part, from these conceptual issues. Habitat selection research is poised to overcome both the major conceptual obstacles and the practical sampling issues that have encumbered its progress. Habitat selection modeling refers generally to quantitative approaches to determine how the physical, chemical, and biological resources and conditions in an area affect occupancy patterns, survival, and reproduction. ‘‘Multi-scale’’ habitat selection modeling refers to any approach that seeks to identify the scale, or scales (in space or time), at which the organism interacts with the environment to determine it being found in, or doing better in, one place (or time) over another. A multi-scale habitat selection study is of great significance for an overall understanding of the ecological habits of animals and the targeted protection and management of wild populations [1]. An increasing number of studies have proven that the solution to an ecological problem depends on the research scale to a large extent, and that different scales may lead to different conclusions [2].

Although the importance of scale has been recognized in ecology, and multi-scale analyses have been gradually advocated, studies on animal habitat selections continue to be limited by scale and the methods of scale analysis. Most studies currently use habitat selection models, where all variables are measured in the same spatial scale, i.e., a single-scale framework for selective analysis, and scale optimization is not considered. Additionally, the choice of scale is often determined by researchers subjectively or based on their knowledge of the ecologies of species [3,4,5,6]. Selecting a univariate model with all variables measured on the same scale would oversimplify the response of species to environmental variables, reduce the interpreted rate of bias, and the predictive power of the model [7,8]. To reduce the effects of bias on the results and improve the performance of the model, researchers should consider using a multi-scale model instead of a single-scale model when evaluating habitat selection. At present, scale analysis encounters two major issues, namely, the selection of an appropriate scale and the transformation and interpretation between scales. The two are closely related and complement each other. The selection or identification of the appropriate scale for a particular problem is a focus of future research in ecology and other disciplines.

As one of the Class Ⅱ species in the list of endangered and protected species of China, red deer (*Cervus elaphus*) is the subject of research in this study. Red deer is second only to moose as a large deer. Red deer live in alpine forests or grasslands and like to live in groups. They do summer activities in the night and early morning and winter activities in the daytime. We used a habitat selection model with a multi-scale framework to study the habitat selection of red deer, and the optimization of the scale of the model. Based on the multi-scale habitat selection model, a suitable habitat for red deer was predicted to provide a reference for a strategy to conserve and recover red deer and its habitat.

## 2. Natural Profile of the Study Area

The Gogostaihanwula Nature Reserve is located in the southern foothills of Greater Khingan, north of Ar Horqin Qiin Chifeng City, Inner Mongolia (44°41′–45°08′ N, 119°03′−119°39′ E). with a total area of 106,284 hectares. The reserve has a semi-arid continental monsoon climate with an average annual temperature of 3.8 °C. The reserve has an altitude of 800–1500 m, and is located in the middle of the eastern section of the southern Greater Khingan Mountains, at the junction of the fauna of the northeast, North China, and New Mongolia, which is a transition zone of forests and grasslands and is the region where the boreal temperate coniferous forest and the East Asian broadleaved forest converge. Due to its unique geographical location, the reserve is rich in biological species and diverse ecosystems.

## 3. Method

### 3.1. Field Data Sampling

We used line transect sampling. We established 75 line transects (5 km long each). 75 transects were placed in the Gogostaihanwula Nature Reserve (at intervals 2 km away from each transect). On each transect large quadrat (10 × 10 m) were set at 500 m intervals. We collected fresh footprints of red deer and environment variable in the large quadrat along the whole length of each transect (5 km). Transects were visited every month between November and the following year in March from 2015 to March 2017. We found 350 presence points of red deer (presence-only data: *n* = 350).

### 3.2. GIS Data and Environmental Variables

The environmental variables affecting the selection of feeding and sleeping habitats of red deer were considered for the study, including deciduous broad-leaved forest ratios, distances from all roads, forest edges, rivers, villages, and cement roads, and the road densities, altitudes, slopes, standard deviations of altitudes, proportions of farmlands and grasslands, and the net primary productivity (see Table 1). All variables were resampled to obtain a uniform spatial resolution of 100 m.

### 3.3. Selection of Appearance Point and Pseudo-Absence Point

Selection of appearance point: Based on a spatial autocorrelation [9], for modeling, this study defines the appearance point of red deer as a distance of more than 1 km, implemented through SDMtoolbox, which is a plug-in of the Geographic Information System (ARCGIS 10.3, https://developers.arcgis.com/) [10]. According to the results of the current studies, the average home range of the red deer is 1 km^2^ [11].

Selection of pseudo-absence points:

Because the absence of red deer during the study period cannot be confirmed as presence or absence of red deer, all red deer presence data should be considered as presence only data. We generated an equal number of random pseudo-absence points to be employed in logistic regression models following a standardized set of procedures. Specifically, for each absence data point, we extracted both the elevation and the distance from the closest road. We then buffered the range of observed elevations by 10% of the difference between the minimum and maximum observed elevations to define the mask of available elevation cells throughout the study area. Next, we calculated the frequency of red deer locations in each 100 m interval distance-from-road bin for the observed data, and we randomly sampled an equal number of pseudo-absence points in each distance bin from the elevation mask. The resulting dataset comprised the pseudo-absence locations dataset.

### 3.4. Model Establishment

The habitat selection model of red deer is at the second level as proposed by Johnson [12], which is the selection of the internal domain of geographical distribution. A multi-scale model was established for the habitat selection of red deer, and logistic regression was used to fit each scale. For the multi-scale model, the pseudo-optimal method was used to establish the multi-scale resource selection function model [13].

Five different scales were calculated for each variable of the appearance point and the available point (the neighborhood edge lengths were 200, 400, 800, 1600, and 3200 m). For land cover type variables, we used focused statistical tools to calculate the proportions of the areas mentioned above across different scale ranges. To determine the means of continuous variables such as altitude and slope within the environmental variables, the distance and other non-scale variables remained unchanged within the scale. The uniform weight of Euclidean distances was used to calculate the values at five different scales for each variable of the appearance and available points. The tool used for calculations was the kernel2dsmooth function [14] in the Smoothie package in the R language (http://www.Rproject.org), for variables in the appearance and available points, using a single-variable logistic regression model and the lowest value of the AICc [15] model as the best scale of the variable. The Pearson coefficient and variable variance inflation factor (VIF) were calculated for the variables, ensuring that the VIF values of each variable were less than 10. All reserved variables of the optimized scale were included in a multivariable logistic regression model.

The model selection method adopted a dredge function in the MuMIn package (MuMIn v1.40.0, https://www.rdocumentation.org/) [16] and selected the optimal model using AICc. If the difference between the AICc minimum model and the AICc second small model was greater than two, then the AICc minimum model was selected as the optimal model [15]. If ΔAICc of several models were all less than two, then a single model could not be selected as the optimal model [15]. In such cases, model averaging is required. The model AIC value less than 2 was selected for averaging. The models with the cumulative weight of the model reaching 90% were selected for average, and the importance score of the variable was obtained by adding the weights of all models with a certain variable and finally using these variables to predict the distribution of red deer in the study area and obtaining a prediction chart as output.

All variables were standardized, and data extraction and analysis were conducted in the ARCGIS 10.3 and R language environments [17].

## 4. Results and Analysis

### 4.1. Multi-Scale Habitat Selection Model

The optimized scales identified by the single-variable logistic regression model showed differences between the variables (see Table 2). Here, the optimal scale of a single variable is mainly explained as follows: In the optimal scale of a single variable, when the circle with the center of 100 m and the radius of the optimal scale the single covariate logistic regression model produces the highest model performance for each variable.

Univariate logistic regression analysis showed that the response scale of red deer to environmental factors was different among different covariates. The optimal scale of the single covariate was 800–3200 m; slope (SLP), altitude (ELE), and ratio of deciduous broad-leaved forests were at a course scale of 800 m, except that the farmland ratio was at a fine scale of 200 m in fine. The optimal scale of road density and grassland ratio is both at 1600 m, and the optimal scale of net forest production capacity is at 3200 m. The Pearson correlation coefficient was used to study the correlations between the variables of the optimized scale (see Figure 1). The correlation coefficient between road density and distance from all roads is −0.72, which is very significant. The correlation coefficient between the ratio of grassland and deciduous broad-leaved forest is −0.87. The correlation coefficient between net primary productivity and distance from forest edges is −0.72. The correlation coefficient between deciduous broad-leaved forest and altitude is −0.71. The variables of smaller AICc are retained. Therefore, four variables including road density, ratio of deciduous forests, ratio of districts, and net primary productivity were not considered. Using the collinearity test of variables with the VIF function, it was found that the VIF value of all variables is approximately two. Therefore, the remaining nine variables were included in the multivariable logistic regression model for further analysis.

Nine variables were considered for modeling. There were no models with ΔAICc < 2. Therefore, we could not determine an optimal model using the AICc. Further, the dredge function of the MuMIn package was used to obtain an average of the models whose cumulative weights were 90%, and the weighting coefficients of the nine variables were obtained for all models (see Table 3).

Based on the importance index of each variable in Table 3, we conclude that the primary factors affecting the habitat selection of red deer are distance from forest edges, distance to cement roads, distance to villages, altitude, distance to all roads, and slope.

The habitat selection of red deer has a positive correlation with the distance to villages and the distance to all roads, and a negative correlation with distance from forest edges, distance to cement road quadratic (It indicates that the red deer choose the appropriate distance from the paved road, first increases with the increase of distance to cement road and then decreases after reaching a peak) altitudes, and slope, indicating that red deer prefer regions with lower altitudes, less interference, gentle slope, and abundant food resources in winter.

The model combinations that affect the habitat selection of red deer (i.e., the optimal model combined with a filter of ΔAICc < 2, as in Table 4) contain a subset of the factors: distance from forest edges, distance to cement roads, distance to villages, altitude, and other covariances.

### 4.2. Prediction of the Suitability of the Red Deer Habitat

Predicted red deer habitat suitability surface for as subarea of the entire study area using model average, and the predictive power of the AUC value of the model was 0.887. Thus, the winter habitat of red deer was determined as having an area of 432.68 km^2^ within a total area of km^2^ (see Figure 2).

## 5. Discussion

### 5.1. Anthropogenic Variables

Results from the multi-scale habitat selection modeling indicate that red deer habitat selection was most strongly related to distance to cement roads, distance to villages, distance to all roads, distance from forest edges, altitude, and slope. Distance to cement roads, distance to villages, and distance to all road: These three variables belong to the anthropogenic variables. The strong (and positive) relationship is consistent with distance to villages and distance to all roads, other studies assessing habitat selection by red deer, and it has been suggested that these red deer select areas with less human interference. Roads are possibly the human infrastructure with the greatest influence on a wide range of organisms [18,19,20], including ungulates [21]. Indeed, a number of ungulate studies have documented avoidance of roads (caribou (*Rangifer tarandus*); elk (*Cervus elaphus*); moose (*Alces alces*)) [22,23,24,25,26,27]. In the most severe cases, roads may act as barriers inhibiting migration between seasonal ranges, resulting in an effective loss of habitat [28,29], and redistribution of individuals measurable at the population level [24,25]. The variable of distance to cement roads are quadratic in our study, given that the red deer choose the appropriate distance from the paved road, first increasing with the increase of distance to cement road and then decreasing after reaching a peak. It is possible that roads are often located in valley bottoms in flat terrain suitable for shrub, which is attractive forage for grazers such as various species of deer. Moreover, roadside shrub may in itself be nutritionally beneficial [30], resulting in a food-driven attraction towards roads. Understanding animal behavior in relation to road networks is necessary to assess the effect of road development on wildlife and to implement appropriate mitigation measures.

### 5.2. Environmental Variables

Red deer habitat selection was a strong negative correlation with environmental variables of slope, altitude, and distance from forest edge. It is likely primarily indicative of the extensive use by red deer of relatively gentle slope, wide area, more forest throughout the study area, which is readily apparent in predicted surfaces generated from the multi-scale models (Figure 2). The use of these areas may also be an indirect result of previous management activities combined with the area of woodland preferred by red deer was basically maintained up to 1995 level. Our study area is a transition zone with forests, grasslands, sand, and other diverse ecosystems. There is no degradation of forest area, except in the case of large-scale grassland degradation of the reserve. specifically, due to reduced grazing. Therefore, these areas may retain more suitable forest for red deer. This is in line with findings by Miao Yang and Libo Zhang [11,31]. Red deer prefer the areas of low altitude and gentle slope that are closer to the edge of the forest. It has been suggested that these red deer select areas with high food availability and open habitat. Red deer preferred habitats with abundant vegetation coverage to open habitats in winter. Our finding is consistent with Mingming Zhang [32]. On the other hand, there were not predators in the reserve. The movement of red deer is limited only by the depth of the snow and the food availability. This is similar to findings by Andrew M. Allen [33], who found that animals distribute, amongst other things, to acquire resources, to reproduce, and to avoid predators or competition with conspecifics. Other research into red deer (*Cervus elaphus*) habitat selection indicates that the relative use of a habitat changes according to its availability, a process known as functional responses in habitat selection [34]. We found that the red deer chose areas closer to the forest edge. This finding is corroborated by a study by Shaochun Zhou [35], who found that the forest thicket margin had positive marginal effect on the population distribution of red deer; in other words, the activity and quantity distribution of red deer in the forest–shrub margin were higher than those in the adjacent thickets and forests.

### 5.3. Scales and Single-Scale vs. Multi-Scale Red Deer Habitat Models

The optimized scales identified by the single-variable logistic regression model showed differences between the variables, except the ratio of farmland variable optimized scales is at fine scale (200 m); the optimal scales of all the remaining variables are at coarse scales (800, 1600, and 3200 m).

We define ‘‘single scale models’’ in this context as models where all covariates are measured at the same scale. Habitat selection studies often use a single scale approach, including red deer. However, there is increasing evidence that biological, ecological, and geographical processes occur at different spatial scales [36]. Therefore, our study is based on a multi-scale approach to habitat selection of red deer. Taking multiple scales into consideration is necessary in order to accurately describe species–habitat relationships [36], yet multiscale habitat selection studies of ungulate are still uncommon [37].

Our study was the first to do so in a multi-scale framework that allowed covariates to enter models at different spatial scales. In our result, we did not compare the multi-scale model with the single-scale model in terms of model explanatory power alone, because we focus on the application of this new method about multi-scale. In future research, we will conduct research in the following aspects: We will compare results from single- and multi-scale models across several criteria, including (1) covariate effect size, (2) variance decomposition, and (3) model explanatory power.

## 6. Management and Conservation Implications

Using the predicted red deer habitat suitability distribution map, it can be seen that the suitable habitat area of the red deer is less than 50% (Figure 2), mostly concentrated in the north. Our results provide an empirical assessment of multi-scale habitat selection for red deer throughout a substantial and important portion of their current range. These results can be used to help guide extensive management efforts that are currently underway in the region aimed at reducing human interference, continuing to slow the degradation of forests, grasslands, and thickets, and banning livestock grazing.

Consistent with previous studies, red deer habitat selection was positively correlated with food availability [38,39,40,41], indicating a need to retain areas with abundant food to provide suitable habitat in the study area. Because red deer will not be in the region because of the snow density increase and reduce the scope of activities, it also reflects that due to special geographical location in the region, in the winter snow cover period, food richness has been lower and red deer cannot meet the demand of energy, but given the poor adaptability of individual natural reduction, it is good for the whole population viability, and the region’s population base is big; therefore, can only more precipitation in the region in early winter can cause supplementary feeding. Due to the consistent finding across numerous studies of the considerable importance of roads in reserve selection by red deer, experimental work assessing potential threshold effects of avoiding roads with respect to red deer habitat suitability would be extremely valuable for wildlife managers.

## Figures and Tables

**Figure 1 animals-10-02454-f001:**
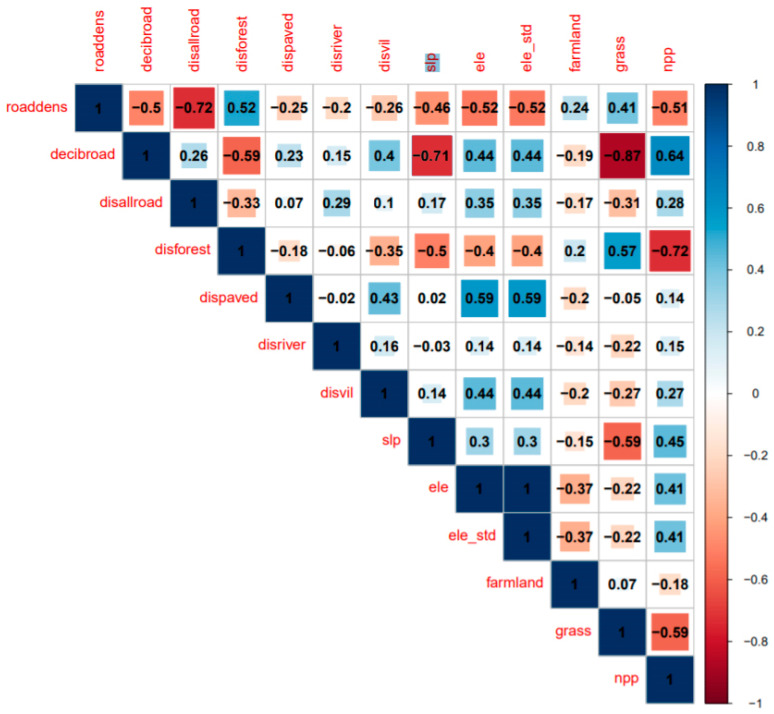
The correlation coefficient matrix between the variables of the optimized scale.

**Figure 2 animals-10-02454-f002:**
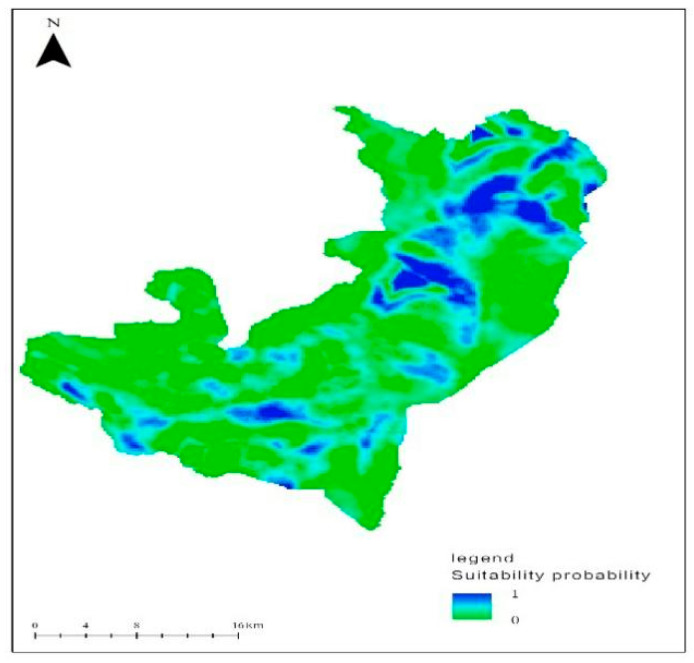
The suitability distribution of the red deer winter habitat in Gogostaihanwula, Inner Mongolia.

**Table 1 animals-10-02454-t001:** Environmental variables used in the analysis of red deer resource selection Forest Service.

Variable	Source	Year
Altitude	Geospatial Data Cloud DEM	2009
Slope	Geospatial Data Cloud DEM	-
Altitude standard deviation	Geospatial Data Cloud DEM	-
Ratio of deciduous broad-leaved forests	Stock map	2004
Ratio of grasslands	Stock map	-
Ratio of farmlands	Stock map	-
Distances to rivers	1:250,000 national basic geographic database	-
Distances to forest edges	1:250,000 national basic geographic database	-
Net primary productivity	MODIS	2015
Distances to villages	1:250,000 national basic geographic database	2015
Road densities	1:250,000 national basic geographic database	-
Distances to all roads	1:250,000 national basic geographic database	-
Distances to cement roads	1:250,000 national basic geographic database	

**Table 2 animals-10-02454-t002:** Level II single variable optimized scale for habitat selection.

Variable	Optimized Scale (m)
Road density (roaddens)	1600
Ratio of deciduous broad-leaved forest (decibroad)	800
Distance to all roads (disallroads)	NA
Distance from forest edge (disforest)	NA (Q)
Distance to cement roads (dispaved)	NA (Q)
Distance to rivers (disriver)	NA
Distance to villages (disvil)	NA (Q)
Slope (slp)	800 (Q)
Altitude (ele)	800 (Q)
Altitude standard deviation (ele_std)	800
Ratio of farmland (farmland)	200
Ratio of grassland (grass)	1600 (Q)
Net primary productivity (npp)	3200

NA refers to a non-scale variable, that is, its value does not change with a change in scale. The letter Q represents a quadratic fit was the best for a given covariate.

**Table 3 animals-10-02454-t003:** Model-averaged interquartile range odds ratios, 95% confidence intervals, and variable importance for the top multi-scale mode.

Variable	Multi-Scale Model	
	Odds Ratio (95% CI)	Importance
intercept	−6.53 (−7.68–−5.39)	1
disforest	−2.60 (−4.47–−0.73)	1
I(dispaved^2)	−2.06 (−3.56–−0.57)	1
disvil	1.26 (0.43–2.08)	1
ele	−1.33 (−2.01–−0.65)	1
disallroad	0.48 (0.04–0.92)	0.92
Slp	−0.59 (−1.21–0.03)	0.91
disriver	0.30 (−0.15–0.76)	0.17
I(slp^2)	−0.24 (−0.73–0.25)	0.11
I(disforest^2)	−2.06 (−7.34–3.23)	0.10
I(ele^2)	−0.23 (−0.94–0.47)	0.08
farmland	−0.19 (−0.89–0.51)	0.07
I(disvil^2)	−0.17 (−1.13–0.79)	0.069
dispaved	0.33 (−2.25–1.59)	0.068
I(farmland^2)	−0.03 (−0.35–0.30)	0.07

**Table 4 animals-10-02454-t004:** The top multi-scale logistic regression models assessing habitat selection by red deer.

Model	D^2^	AICc	ΔAICc	Weight AICc
disallroad + disforest + dispaved* + disvil + ele + slp	7	294.41	0	0.17
disallroad + disforest + dispaved* + disvil + ele + slp + slp*	8	295.37	0.96	0.11
disallroad + disforest + disforest* + dispaved* + disvil + ele + slp+	8	295.54	1.13	0.10
disallroad + disforest + dispaved* + disriver + disvil + ele + slp	8	295.75	1.34	0.09
disallroad + disforest + dispaved* + disvil + ele	6	295.75	1.34	0.09
disforest + dispaved* + disriver + disvil + ele + slp	7	295.92	1.50	0.08
disallroad + disforest + dispaved* + disvil + ele + ele* + slp	8	295.92	1.51	0.08
disallroad + disforest + dispaved* + disvil + ele + farmland + slp	8	296.11	1.70	0.07
disallroad + disforest + dispaved* + disvil + disvil* + ele + slp	8	296.28	1.87	0.07
disallroad + disforest + dispaved + dispaved* + disvil + ele + slp	8	296.29	1.88	0.07
disallroad + disforest + dispaved* + disvil + ele + farmland* + slp	8	296.38	1.97	0.07

* represents a quadratic fit was the best for a given covariate.

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
