# Peer review of "The Winter Habitat Selection of Red Deer (Cervus elaphus) Based on a Multi-Scale Model"

_animals, 2020, doi:10.3390/ani10122454_

Round 1

Reviewer 1 Report

The ms describes habitat selection by red deer, emphasising the importance of scale. Looks like an appropriate field design and analysis for the questions addressed. Generally clearly written. Would benefit from some graphics illustrating the form of preference effects observed.

l41 Ref 13 is also a good source for this point.

L56 Indicate this class of protection applies to China.

L73 Say what the samples were before saying how many there were. Section 3.1 is not clear on either spatial arrangement of the sampling units, or on what was actually counted on the sample lines – ‘10m x 10 m sampling square’ is baffling.

L82 The ‘resampling’ is unclear.

L85 The description of field method is also incomplete here: were human observers walking along transects ? If so, with or without binoculars ?

L88 “The average home range of male red deer at this is approximately 1 km2” Is something distinct implied by ‘domain’ ? Does the cited work give an SE (or CI) for the range ?

L96-99. Does the procedure described here, where the pseudo-absence distribution is simulated so as to have the same distribution as that of the deer locations, not substantially reduce if not remove the possibility of detecting selection with respect to road distance ?

L102 ‘Johnson’ not ‘John’.

L104 Briefly say what is meant by ‘pseudo-optimal’.

L115 Is it intended that the ‘smallest’ AIC corresponds to the ‘optimal’ scale or best-fit scale (not ‘smallest’ scale). The single variable models were also tested for presence of a quadratic effect/ (as noted in table 2 legend)

L116 VIF is Variance Inflation Factor.

L119 What was the total number of models dredged? With 9 variables (assuming an optimal scale is first selected for each), then have ((2^9)-1)= 511 ?

L123 Is this a description of model averaging – ie obtained a single averaged model? Looks like 11 models were within 2.0 AIC of the model with lowest AIC (table 4).

L134 So where correlations were high, one was omitted, and this occurred for 4 variables. How was it decided which to drop ?

Table 3 Too many decimal places here (and in table 4 also). Put main effects and quadratic terms on adjacent rows. Add something to indicate which of these CI do not include zero – currently hard to see.

L156-161 Effects which have quadratic terms are difficult to interpret - some plots of these effects would be useful. Describe the effect – saying preference was related to distance but not how is not helpful. Put the cement road description in a separate sentence.

L158 ‘binomial’ ? – was ‘quadratic intended ? what shape is the curve ? Peaked at an intermediate value it seems – again, a plot would be very useful.

L168 Was the top model used for this ? why not the averaged model?

Discussion

Is it possible to say here exactly how conclusions would have been misleading if scale had not been taken into account?

Second sentence is circular.Third sentence better framed as the multi-scale model being more biologically plausible than any single scale model ?

L183 ‘defenders ‘ ? Conservationists ?

Is it possible to say anything about some specifics – why would ration of farmland be more influential at small scales compared with nnp ? (table 2), for example.

L194 Need only say ‘Red deer appeared to prefer terrain with gentle slopes ‘ , sentence ‘In this study..’   not needed.

What is the siginificance of the final sentence – deer population is increasing, depleting food, so range of individuals expanding, how does that affect observed preference at different scales?

L235 capital ‘G’ in ‘McGarigal’.

Author Response

The ms describes habitat selection by red deer, emphasising the importance of scale. Looks like an appropriate field design and analysis for the questions addressed. Generally clearly written. Would benefit from some graphics illustrating the form of preference effects observed.

l41 Ref 13 is also a good source for this point.

Response: Thank you very much for your affirmation.

L56 Indicate this class of protection applies to China.

Response: There are 8 wild red deer subspecies in my country:Cervus xanthopygus、Cervus elaphus yarkandensis、Cervus elaphus songaricus、Cervus elaphus sibiricus、Cervus elaphus  kansuensis、Cervus elaphus alashanicus、Cervus elaphus wallichi、Cervus elaphus  maacneilli.Their habitats are gradually decreasing,only distributed in HeilongJiang,Jilin, Xinjiang, Gansu, Ningxia, Qinghai, Sichuan, Tibet, Inner Mongolia and Hebei. Due to habitat destruction and illegal hunting, the wild red deer is endangered. The population is small, only 130,000, and its wild conditions are not optimistic.Therefore,the research on the habitat selection of the wild population of red deer is particularly important.Finding the research methods of species habitat selection is very important for restoration of red deer habitat and population protection.

L73 Say what the samples were before saying how many there were. Section 3.1 is not clear on either spatial arrangement of the sampling units, or on what was actually counted on the sample lines – ‘10m x 10 m sampling square’ is baffling.

Response: We used the Line transect sampling. According to the existing topographic forest phase map, We set systematic transect lines at intervals of more than 2 km in the research area. Each transect line is 5km.On each transect line large quadrat (10 m×10 m) were set at 500 m intervals. We set 75 transect lines in total within the Gogostaihanwula Nature Reserve from December 2015 to March 2017. The number of large quadrat incluing red deer presence point was found to be 350(presence-only data: n=350.

L82 The ‘resampling’ is unclear.

Response: All GIS data from multiple sources were used to create gridded environmental variables (100m resolution).In brief,four nonscalar environmental variables were used in each model and all other variables were measured at five scales(200,400,800,1600,3200m) by measuring cell statistics within different sized windows centred on each raster cell using the focal statistics tool in ArcGis spatial analyst tool.

L85 The description of field method is also incomplete here: were human observers walking along transects ? If so, with or without binoculars ?

Response: Without binoculars.

We used the Line transect sampling. According to the existing topographic forest phase map, We set systematic transect lines at intervals of more than 2 km in the research area. Each transect line is 5km.On each transect line large quadrat (10 m×10 m) were set at 500 m intervals. We set 75 transect lines in total within the Gogostaihanwula Nature Reserve from December 2015 to March 2017. The number of large quadrat incluing red deer presence point was found to be 350(presence-only data: n=350).

L88 “The average home range of male red deer at this is approximately 1 km2” Is something distinct implied by ‘domain’ ? Does the cited work give an SE (or CI) for the range ?

Response: 114.58±3.19ha.

L96-99. Does the procedure described here, where the pseudo-absence distribution is simulated so as to have the same distribution as that of the deer locations, not substantially reduce if not remove the possibility of detecting selection with respect to road distance ?

Response: No substantial reduction.

L102 ‘Johnson’ not ‘John’.

Response: Yes, Clerical error. Thanks.

L104 Briefly say what is meant by ‘pseudo-optimal’.

Response: In the most common version of this approach, the investigator evaluated each covariate separately across a range of pre-specified scales and used statistical measures to select the single best scale for each covariate, and then combined the covariates (at their best univariate scale) into some average multi-variable, multi-scale model. Note, we refer to this approach as ‘‘pseudo-optimized’’.

L115 Is it intended that the ‘smallest’ AIC corresponds to the ‘optimal’ scale or best-fit scale (not ‘smallest’ scale). The single variable models were also tested for presence of a quadratic effect/ (as noted in table 2 legend)

Response: Yes, It is. The optimal scale of each variable is determined by the one-to-one corresponding AICc value of 5 scales (200, 400, 800, 1600, 3200). The smallest AICc value is its optimal scale. Univariate quadratic terms are also included in the model.

L116 VIF is Variance Inflation Factor.

Response:Thanks, I have modified.

L119 What was the total number of models dredged? With 9 variables (assuming an optimal scale is first selected for each), then have ((2^9)-1)= 511 ?

Response: 511.

L123 Is this a description of model averaging – ie obtained a single averaged model? Looks like 11 models were within 2.0 AIC of the model with lowest AIC (table 4).

Response: It is not a single averaged model.If the difference between the smallest AICc model and the second smallest model of AICc is greater than 2,then the model with the smallest AICc is selected as the most optimal model; if there are several models with ΔAICc less than 2, then no single model can be selected as Optimal model. At this time, model averaging is required.

L134 So where correlations were high, one was omitted, and this occurred for 4 variables. How was it decided which to drop ?

Response: If the absolute value of the correlation coefficient between variables is greater than 0.7,retain the variable with a small AICc value.

Table 3 Too many decimal places here (and in table 4 also). Put main effects and quadratic terms on adjacent rows. Add something to indicate which of these CI do not include zero – currently hard to see.

Response: The two decimal places are uniformly reserved, and the original text has been revised. None of these CI's in our results are 0.

L156-161 Effects which have quadratic terms are difficult to interpret - some plots of these effects would be useful. Describe the effect – saying preference was related to distance but not how is not helpful. Put the cement road description in a separate sentence.

Response: I understand what you mean, and thank you very much for your comments.I have made changes in the original text.

L158 ‘binomial’ ? – was ‘quadratic intended ? what shape is the curve ? Peaked at an intermediate value it seems – again, a plot would be very useful.

Response: Wrong writing, it should be a quadratic term. In the multi-scale model, the habitat selection of red deer is negatively correlated with the quadratic term of the variable distance to the cement road, so the parabola opens downward and has a peak.

L168 Was the top model used for this ? why not the averaged model?

Response:The result is made with the average model. I only listed 11 multi-scale models with ΔAICc less than 2.

Discussion

Is it possible to say here exactly how conclusions would have been misleading if scale had not been taken into account?

Response: Organisms select habitat at multiple hierarchical levels and at different spatial and/or temporal scales within each level.The environment is structured across scales in space and time, and organisms perceive and respond to this structure at different scales.Indeed, how environmental structure affects the distribution, abundance and fitness of organisms has been the focus of ecology since its inception. In the past researchers studying red deer habitat selection process, the scales of all variables are fixed, that is, the size of the 200-meter grid is very subjective, and the scale of the selection of variable factors by animals should not be fixed. The concept that species respond to their environment across a range of spatial scales, both within and among specific habitat components, has long been appreciated in ecology, though the implementation of this concept into habitat selection studies has lagged considerably.

Second sentence is circular.Third sentence better framed as the multi-scale model being more biologically plausible than any single scale model ?

Response: I have modified in the original.

L183 ‘defenders ‘ ? Conservationists ?

Response: I have modified in the original.

Is it possible to say anything about some specifics – why would ration of farmland be more influential at small scales compared with nnp ? (table 2), for example.

Response: Because the farmland is covered by heavy snow in winter, there is no farmland that can satisfy the survival of red deer.Structural resources such as concealment and food have caused the red deer not to choose habitats such as farmland in winter, and deliberately avoid farmland, and choose the scale 200m away from farmland as the optimal scale. Forest net primary productivity (npp) is the carbon cycle process of the ecosystem. An important part of the project, the scale is larger, and it is more direct without the impact of farmland.

L194 Need only say ‘Red deer appeared to prefer terrain with gentle slopes ‘ , sentence ‘In this study..’   not needed.

Response: Thanks, I have modified.

What is the siginificance of the final sentence – deer population is increasing, depleting food, so range of individuals expanding, how does that affect observed preference at different scales?

Response: I did not mention the increase in the number of red deer in the original text. In order to obtain more food, the home area of red deer has increased. The main factors reflect the main factors that affect the food choices of red deer in winter. These factors may be related to red deer getting food.

L235 capital ‘G’ in ‘McGarigal’.

Response: Thanks, I have modified.

Reviewer 2 Report

This is an interesting paper using modelling to predict habitat for red deer in Mongolia. Overall the modelling is sound and the paper could have an impact on deer conservation or a better understanding at least of their resource use in an important area. Please find some comments that I hope help you improve the paper

The abstract needs methods and some numerical results as well as a final statement to make the work more broadly applicable.

Line 56 - a bit more about red deer as a model, including its scientific name

Line 74 - it is confusing how the sampling was done - was it on foot? with cameras? by road?

It is also not clear how random the sampling was as the sampling seems to have been done in areas WITH red deer; can you be more clear about random plots? In this way, the ideal habitat predictions would make more sense, as otherwise you pre-determined it by choice of transects.

You have a lot of variables in the analysis - even if they were not all autocorrelated, did you get any better models with a more variable delta AIC by excluding some variables?

The weakest part of this paper is the discussion - only three other studies are cited - this paper would be much more powerful if it was put better in the context of the literature.

Author Response

This is an interesting paper using modelling to predict habitat for red deer in Mongolia. Overall the modelling is sound and the paper could have an impact on deer conservation or a better understanding at least of their resource use in an important area. Please find some comments that I hope help you improve the paper.

The abstract needs methods and some numerical results as well as a final statement to make the work more broadly applicable.

Response: I have adopted your advise and I have revised the abstract.

Line 56 - a bit more about red deer as a model, including its scientific name

Response: I have made changes in the introduction.

Line 74 - it is confusing how the sampling was done - was it on foot? with cameras? by road?

Response: Walking, without camera, using line transect, the sample line is generally laid perpendicular to the road.

It is also not clear how random the sampling was as the sampling seems to have been done in areas WITH red deer; can you be more clear about random plots? In this way, the ideal habitat predictions would make more sense, as otherwise you pre-determined it by choice of transects.

Response: We used the Line transect sampling. According to the existing topographic forest phase map, We set systematic transect lines at intervals of more than 2 km in the research area. Each transect line is 5km.On each transect line large quadrat (10 m×10 m) were set at 500 m intervals. We set 75 transect lines in total within the Gogostaihanwula Nature Reserve from December 2015 to March 2017. The number of large quadrat incluing red deer presence point was found to be 350(presence-only data: n=350).

You have a lot of variables in the analysis - even if they were not all autocorrelated, did you get any better models with a more variable delta AIC by excluding some variables?

Response: These variables have been shown to be important factors in previous studies and should not be eliminated when building the model.

The weakest part of this paper is the discussion - only three other studies are cited - this paper would be much more powerful if it was put better in the context of the literature.

Response: Thanks for the valuable comments, I have revised the discussion, please review it again.

Reviewer 3 Report

The authors applied a relatively novel technique to determine the habitat selection of red deers. While the methods and results are relatively clear and potentially interesting, the introduction and discussion are very weak. I recommend working on the following major issues:

  • There should be a general introduction on habitat selection, now completely missing, before going into multi-scale habitat selection
  • You say your aim is to use your multi-scale approach to provide a reference study for the conservation of red deer, but it is not clear how. Also, you should introduce examples of model selection in similar animals to make predictions on the variable you are testing. And explain why you want to apply a multi-scale approach and what are your expectations.
  • It is not clear why red deers selected the different variables at different scales. It is not thus clear why you are using a multi-scale approach if you do not discuss the implications of it. The interpretation of the results is very poor, and it is not clear how your findings are linked to the current literature and how your findings can help conservationists and ecologists. 

Other small issues:

  • Remove "A Study of The" from the title
  • Species name should not be capitalised, same in the simple summary.
  • The abstract and simple summary should be changed taking into account the major issues
  • line 73, Samples of what? What did you collect?
  • What do you mean by "All variables were resampled to obtain a uniform spatial resolution of 100"? What is the real resolution of each layer you used? 
  • Table 1. Why you do not have the year for all the variables? It is important to add for the layers Ratio of grasslands and Ratio of farmlands, but also distance from roads, as they might change.
  • Line 107. Why did you select those scales? It is not clear to me the criterion used
  • Lines 137-138. Why you did not consider those variables? Need to explain.
  • Table 2. You say 9 variables were included in the model and 4 excluded, but from this table (and also from table 4) it seems you included 8 variables. Please recheck your variable selection.
  • Table 3. If possible, it would be nice to have significant relationships in a figure so that they are clearer. See the following paper for an example Klaassen B, Broekhuis F. Living on the edge: Multiscale habitat selection by cheetahs in a human‐wildlife landscape. Ecol Evol. 2018;8:7611–7623.
  • Figure 2. The figure text should be in English and the figure should be at high resolution. 

Author Response

The authors applied a relatively novel technique to determine the habitat selection of red deers. While the methods and results are relatively clear and potentially interesting, the introduction and discussion are very weak. I recommend working on the following major issues:

There should be a general introduction on habitat selection, now completely missing, before going into multi-scale habitat selection

Response: Yes, I have added in the original text.

First, ‘‘habitat selection modeling’’refers generally to quantitative approaches to determine how the physical, chemical and biological resources and conditions in an area affect occupancy patterns, survival and reproduction. In this context,‘‘multi-scale’’ habitat selection modeling refers to any approach that seeks to identify the scale, or scales (in space or time), at which the organism interacts with the environment to determine it being found in, or doing better in, one place (or time) over another.

You say your aim is to use your multi-scale approach to provide a reference study for the conservation of red deer, but it is not clear how. Also, you should introduce examples of model selection in similar animals to make predictions on the variable you are testing. And explain why you want to apply a multi-scale approach and what are your expectations.

It is not clear why red deers selected the different variables at different scales. It is not thus clear why you are using a multi-scale approach if you do not discuss the implications of it. The interpretation of the results is very poor, and it is not clear how your findings are linked to the current literature and how your findings can help conservationists and ecologists. 

Response: Determining the best scale(s) at which to describe habitat selection is a major focus of current multi-scale habitat selection modeling.

Habitat selection is the disproportionate use of available conditions and resources, and involves responses in space and time to perceived risks and rewards. It frequently depends on the scale of measurement, often in non-linear ways that preclude simple extrapolation across scales. More critically, animals often select different habitat components at different scales, and species vary in their scales of selection.

Based on the multi-scale research method, this article derives the main factors affecting the habitat selection of red deer and provides guidance for protecting the habitat of red deer. However, the multiscale approach within habitat selection studies is still uncommonand to our knowledge this is the first multiscale habitat selection study for red deer.

At present, no such multi-scale method has been used in the selection of deer habitats.

First, ‘‘habitat selection modeling’’ refers generally to quantitative approaches to determine how the physical, chemical and biological resources and conditions in an area affect occupancy patterns, survival and reproduction. In this context, ‘‘multi-scale’’ habitat selection modeling refers to any approach that seeks to identify the scale, or scales (in space or time), at which the organism interacts with the environment to determine it being found in, or doing better in, one place (or time) over another.

My expectation is to more accurately reveal the important factors that influence the selection of red deer habitat, and to provide assistance to environmentalists and ecologists in restoring the red deer habitat and protecting the population.The investigator pre-selected one scale (Generally 100m)for all covariates within each level,the investigator evaluated each covariate separately across a range of pre-specified scales and used statistical measures to select the single best scale for each covariate, and then combined the covariates (at their best univariate scale) into a single multi-variable,multiscale model.Importantly,in this approach the investigator used empirical means to determine the best scale for each variable, but the investigator also allowed the variables to enter the multi-variable model at different scales. Hence, the final model in this approach was truly multi-scale.

Other small issues:

Remove "A Study of The" from the title

Response: Thanks for your suggestion, I have revised the title.

Species name should not be capitalised, same in the simple summary.

Response: Thanks, I have modified in the original text.

The abstract and simple summary should be changed taking into account the major issues

line 73, Samples of what? What did you collect?

Response: Thanks, I have modified in the original text.

We used the Line transect sampling. According to the existing topographic forest phase map, We set systematic transect lines at intervals of more than 2 km in the research area. Each transect line is 5km.On each transect line large quadrat (10 m×10 m) were set at 500 m intervals. We set 75 transect lines in total within the Gogostaihanwula Nature Reserve from December 2015 to March 2017. The number of large quadrat incluing red deer presence point was found to be 350(presence-only data: n=350).

What do you mean by "All variables were resampled to obtain a uniform spatial resolution of 100"? What is the real resolution of each layer you used?

Response: All GIS data from multiple sources were used to create gridded environmental variables (100m resolution). In brief, four non-scalar environmental variables were used in each model and all other variables were measured at five scales (200,400,800,1600,3200m) by measuring cell statistics within different sized windows centred on each raster cell using the focal statistics tool in ArcGis spatial analyst tool.

The spatial resolution of each variable is base on 100m*100m in GIS.

Table 1. Why you do not have the year for all the variables? It is important to add for the layers Ratio of grasslands and Ratio of farmlands, but also distance from roads, as they might change.

Response: We sampled for three consecutive years, ignoring inter-annual changes, and put together the red deer occurrence points found each year for common statistical analysis. At the same time, we cannot obtain the data of the annual increase in Ratio of grasslands and Ratio of farmlands.

Line 107. Why did you select those scales? It is not clear to me the criterion used

Response: The smallest scale of 200m is close to the spatial resolution grid size,and the largest scale of 3200m includes the home area of the red deer,that is,the maximum moving distance.

Lines 137-138. Why you did not consider those variables? Need to explain.

Response: If the absolute value of the correlation coefficient between variables is greater than 0.7, keep the variable with a small AICc value.

Table 2. You say 9 variables were included in the model and 4 excluded, but from this table (and also from table 4) it seems you included 8 variables. Please recheck your variable selection.

Response: There are 9 variables, resulting in a total of 2^9)-1)=511 models. The models with △AICc less than 2 are listed in Table 4. The other models are not listed.

Table 3. If possible, it would be nice to have significant relationships in a figure so that they are clearer. See the following paper for an example Klaassen B, Broekhuis F. Living on the edge: Multiscale habitat selection by cheetahs in a human‐wildlife landscape. Ecol Evol. 2018;8:7611–7623.

Response: Thank you very much for your valuable suggestions.

Figure 2. The figure text should be in English and the figure should be at high resolution. 

Response: Thanks, I have modified in the original text.

Round 2

Reviewer 2 Report

I have read the revised version of Sun et al's habitat selection of red deer and applaud the authors on many changes that I requested in the first version. Thank you for including more methods and results in the abstract. This really helps the reader. I still suggest to make the final sentence of the abstract more broad as more people will read this article beyond those interested in red deer.

Remember also that measures and numbers must be separated - e.g. 3200 m (because the measure is a new word).

I am curious why there are only three key words - again broadening this would allow more readers to find your paper

The first paragraph is now more detailed but now very long. Probably around line 59 you can make a new paragraph.

Line 74 - please explain and cite what you mean by second class - this must be in Chinese law? Make it sow the international reader can understand. There are some notes in this section that are not complete sentences (e.g. like to live in groups) - please expand and give citations.

Line 93 - this section is improved as we needed the method, but it is full of mistakes - please get a native English speaker to check. the Line transect sampling method has many variations - you need to say exactly what you did and give the citation.

Line 148 - thank you for clarifying the AIC

Line 199 -  something is missing from this sentence - were these two models or one?

Line 220 - this idea of multiple for Ungulata is really exciting and is the kind of wording needed to broaden the last line of the abstract (and could be help with the key words)

232-233 I think you mean increasing when further distant from the cement road and decreasing after reaching a peak - the independent variable sentence is not needed

It is still an incredibly short discussion but maybe that is the style of this type of contribution to the journal? I would expect the results also to consider at least a paragraph or two of the last research aims -those being discussion "a suitable habitat for red deer was predicted to provide a reference for a strategy to conserve and recover red deer and its habitat."

Author Response

Remember also that measures and numbers must be separated - e.g. 3200 m (because the measure is a new word).

Response: Thank you very much for your valuable suggestion. I have revised it.

I am curious why there are only three key words - again broadening this would allow more readers to find your paper

Response: Thank you very much for your valuable advice. I added a few key words.

The first paragraph is now more detailed but now very long. Probably around line 59 you can make a new paragraph.

Response: I have revised it.

Line 74 - please explain and cite what you mean by second class - this must be in Chinese law? Make it sow the international reader can understand. There are some notes in this section that are not complete sentences (e.g. like to live in groups) - please expand and give citations.

Response: Thank you very much for your valuable advice. I have revised it.

Line 93 - this section is improved as we needed the method, but it is full of mistakes - please get a native English speaker to check. the Line transect sampling method has many variations - you need to say exactly what you did and give the citation.

Response: We established 75 line transects(5 km long each). 75 transects were placed in the the Gogostaihanwula Nature Reserve (at intervals 2 km away from each transect). On each transect large quadrat (10 m×10 m) were set at 500 m intervals. We collected fresh footprints of red deer and environment variable in the large quadrat along the whole length of each transect (5 km).Transects were visited every month between November and the following year in March from 2015 to March 2017. We found 350 presence points of red deer.

Line 148 - thank you for clarifying the AIC

Response: Thank you for your meticulous revision of my paper.

Line 199 -  something is missing from this sentence - were these two models or one?

Response: There are many models,and make these models average.So it is not one model or two models.

Line 220 - this idea of multiple for Ungulata is really exciting and is the kind of wording needed to broaden the last line of the abstract (and could be help with the key words)

Response: Thank you for your meticulous revision of my paper. I have revised it

232-233 I think you mean increasing when further distant from the cement road and decreasing after reaching a peak - the independent variable sentence is not needed

Response: Thank you for your meticulous revision of my paper. I have revised it

It is still an incredibly short discussion but maybe that is the style of this type of contribution to the journal? I would expect the results also to consider at least a paragraph or two of the last research aims -those being discussion "a suitable habitat for red deer was predicted to provide a reference for a strategy to conserve and recover red deer and its habitat."

Response: I would like to thank you for your attention to my paper and all kinds of valuable opinions. I have revised them carefully.In the future scientific research work, I will certainly carry out scientific research in accordance with your valuable suggestions that are helpful for me to write my thesis. Once again, I sincerely thank you!Hope you have a good mood and good health every day!

Reviewer 3 Report

The paper is overall improved although there are several things to fix before publication. There are many tiny mistakes in style, formatting, etc. so I suggest deep proofreading before resubmitting the next revisions. For example, i) it is still Cervus Elaphus instead of Cervus elaphus in the title and simple summary; ii) there are many capital letters in the middle of the sentences. Then there is a phrase repeated twice at lines 229-230. Please make sure all these tiny mistakes and others are fixed. 

I still find unclear why you used a multi-scale approach as you do not really discuss this aspect in your discussion. For example, what does it mean that road density optimised scale is at 1600 m? what does it mean that the optimised Ratio of deciduous broad-leaved forest is at 800 m? And so on, all the results in table 2 are not explained. Take for example the paper I suggested in my previous review for an example on how to discuss your findings. Klaassen B, Broekhuis F. Living on the edge: Multiscale habitat selection by cheetahs in a human‐wildlife landscape. Ecol Evol. 2018;8:7611–7623. Also, in that paper, you find some additional ways to show your results that I suggested in the last review but you did not consider. That can ameliorate the paper. I still think that this paper has good potential but it is not fully explored. You often justify your paper and findings saying they are the first application of a multi-scale approach to red deer, but I expect you to refer to research on other species as well.   

Author Response

Comments and Suggestions for Authors

The paper is overall improved although there are several things to fix before publication. There are many tiny mistakes in style, formatting, etc. so I suggest deep proofreading before resubmitting the next revisions. For example, i) it is still Cervus Elaphus instead of Cervus elaphus in the title and simple summary; ii) there are many capital letters in the middle of the sentences. Then there is a phrase repeated twice at lines 229-230. Please make sure all these tiny mistakes and others are fixed.

Response: Thank you very much for your valuable suggestion. I have revised it.

I still find unclear why you used a multi-scale approach as you do not really discuss this aspect in your discussion.

For example, what does it mean that road density optimised scale is at 1600 m? what does it mean that the optimised Ratio of deciduous broad-leaved forest is at 800 m? And so on, all the results in table 2 are not explained.

Response: Thank you very much for your valuable suggestion.

We employed standard logistic regression to develop a Resource Selection Function (RSF) of red deer habitat selection within the study area. We first conducted a univariate scaling analysis to empirically identify and select the characteristic scale and the functional form (i.e., standard logistic or quadratic logistic) combination for each covariate to be used in multi-scale models. We then employed each of these covariates in a single covariate logistic regression model using standard logistic and quadratic logistic functional forms independently, and we retained the scale and functional form combination with the lowest Akaikes Information Criterion corrected for small sample size (AICc) value among all candidates.

Specifically, for each covariate we calculated the uniform kernel density value (i.e., the uniform weighted mean as a function of Euclidean distance) across a range of bandwidths extending at five scale (200m,400m,800m,1600m,3200m) for each use and pseudo-absence location.This is completely different from the previous research methods of habitat selection in which variable attributes were extracted at a fixed scale of 100m or 200m.

Prior to running multiple logistic regression analyses, we calculated Pearsons correlations among all covariates to assess potential multicollinearity. In instances of high pairwise correlation between covariates (i.e., |r|0.7), we retained the covariate with the greater deviance explained and removed the other covariate from subsequent analyses. We calculated variance inflation factor (VIF) for all remaining covariates and confirmed that none of the retained covariates had a VIF10.

We then used AICc and Akaike’s model weights (x) to rank the candidate models and select the model(s) that best separated red deer use locations from pseudo-absence locations. We considered models2 AICc units from the best supported model to be jointly supported (Burnham and Anderson 2002). If no single model comprised90 % of the weight of the entire candidate model set, we used model averaging to derive parameter estimates from the top models that accounted for90% of the cumulative model weights(Burnham and Anderson 2002)

For example the explanation of the optimal scale for the variable of road density at 1600m: For the variable of road density, when the road density within the radius of 1600m with the 100-meter grid as the center the single covariate logistic regression model produces the maximum model predictive performance at this scale.

For example the explanation of the optimal scale for the variable of Ratio of deciduous broad-leaved forest at 800m: For the variable of ratio of deciduous broad-leaved forest, when the ratio of deciduous broad-leaved forest within the radius of 800m with the 100-meter grid as the center the single covariate logistic regression model produces the maximum model predictive performance at this scale.

Take for example the paper I suggested in my previous review for an example on how to discuss your findings. Klaassen B, Broekhuis F. Living on the edge: Multiscale habitat selection by cheetahs in a human‐wildlife landscape. Ecol Evol. 2018;8:7611–7623.

Response: Thank you very much for your valuable suggestion. I have revised my discussion.

Also, in that paper, you find some additional ways to show your results that I suggested in the last review but you did not consider. That can ameliorate the paper. I still think that this paper has good potential but it is not fully explored. You often justify your paper and findings saying they are the first application of a multi-scale approach to red deer, but I expect you to refer to research on other species as well.  

Response: I would like to thank you for your attention to my paper and all kinds of valuable opinions. I have revised them carefully.In the future scientific research work, I will certainly carry out scientific research in accordance with your valuable suggestions that are helpful for me to write my thesis. Once again, I sincerely thank you!Hope you have a good mood and good health every day!